

# Empirical model for backscattering polarimetric variables in rain at W-band: motivation and implications

Alexander Myagkov[1], Tatiana Nomokonova[1], and Michael Frech[2]

[1]Radiometer Physics GmbH, Meckenheim, Germany
[2]Observatorium Hohenpeißenberg, Deutscher Wetterdienst, Hohenpeißenberg, Germany

**Correspondence:** Alexander Myagkov (alexander.myagkov@radiometer-physics.de)

**Abstract.**

The established relationships between the size, shape, and terminal velocity of raindrops, along with the spheroidal shape approximation (SSA), are commonly employed for calculating radar observables in rain. This study, however, reveals the SSA's limitations in accurately simulating spectral and integrated backscattering polarimetric variables in rain at the W-band.

Improving existing models is a complex task that demands high-precision data from both laboratory settings and natural rain, enhanced stochastic shape approximation techniques, and comprehensive scattering simulations. To circumvent these challenges, this study introduces a simpler and more straightforward approach – the empirical scattering model (ESM).

The ESM is derived from an analysis of high-quality, low-turbulence Doppler spectra, which were selected from measurements taken with a 94 GHz radar at three different locations between 2021 and 2024. The ESM's primary advantages over the

SSA include superior accuracy and the direct incorporation of microphysical effects observed in natural rain.

This study demonstrates that the ESM can potentially clarify issues in existing retrieval and calibration methods that use polarimetric observations at the W-band. The findings of this study are not only valuable for experts in cloud radar polarimetry but also for scattering modelers and laboratory experimenters since explaining the presented observations necessitates a more profound understanding of the microphysical properties and processes in rain.

## 15  1  Introduction

Over the past few decades, numerous theoretical and empirical studies have been conducted to explore the relationship between the size, shape, and terminal velocity of water droplets. A comprehensive review of the history and techniques of drop measurement can be found in Kathiravelu et al. (2016). Early studies, mostly based on laboratory measurements (e.g. Laws, 1941; Gunn and Kinzer, 1949; Best, 1950; Medhurst, 1965; Foote and Toit, 1969; Pruppacher and Pitter, 1971; Beard, 1976; Beard

and Chuang, 1987), assign average terminal velocity and axis ratio to a given drop size. Even though these approximations do not take into account a number of effects occurring in natural rain, simplicity and a tolerable accuracy of these approximations motivate their wide utilization. Later studies (e.g. Thurai and Bringi, 2005; Thurai et al., 2007, 2021) have been focused on natural rain and are often based on a careful processing of a large datasets from 2D video disdrometers (Kruger and Krajewski, 2002).



The interest in the characteristics of individual water drops stems from the importance of rain microphysics for precipitation-oriented applications in meteorology, hydrology, and agriculture. The size-velocity relation establishes a link between a drop-size distribution (DSD) and widely used integral rain properties such as intensity, accumulated amount, and kinetic energy. The relation between size and shape influences the propagation and scattering properties of a medium containing raindrops, making this relation vital for telecommunications and precipitation remote sensing, especially when polarimetry is employed

(Oguchi, 1983).

Certain dependencies between drop properties have to be assumed in rain retrievals based on in situ and remote sensing instruments (Löffler-Mang and Joss, 2000; Peters et al., 2002; Matrosov et al., 2002; Ryzhkov et al., 2005b; Kwon et al., 2020; Li et al., 2023). For instance, optical disdrometers assume the relations between drop properties to convert the observed laser-beam attenuation and time period during which a water particle crosses the beam to size and velocity. Vertically pointed

micro rain radars (MRR) use radial velocity of raindrops as a proxy of their size for DSD profiling. Assumed relations between drop size and shape are required for advanced quantitative precipitation estimation, correction for propagation effects, and calibration evaluation in polarimetric centimeter-wavelength radars. A characterization of size-shape-velocity relations is also necessary in forward models (e.g. Cao et al., 2010; Wolfensberger and Berne, 2018; Mahale et al., 2019; Matsui et al., 2019) used for variational retrievals and for evaluation of weather models. Properties of water drops have also been used to explain

spectral radar observations in rain. For instance, Moisseev and Chandrasekar (2007); Tridon and Battaglia (2015) present two DSD-profiling approaches based on spectral polarimetry and dual frequency Doppler spectra, respectively. The selected dependencies might affect retrieval results. This, however, is discussed only in a limited number of studies (Testud et al., 2000; Gorgucci et al., 2006; Thurai et al., 2007; Gorgucci and Baldini, 2009).

Millimeter-wavelength radars (cloud radars hereafter) have become a crucial tool for remote sensing of clouds and precipi-

tation. These instruments have found extensive application across various climatic regions. For example, cloud radars are used to investigate ice containing clouds in the Arctic, liquid clouds in Tropics, thunderstorms in mid-latitudes, etc. A review of cloud radar applications can be found in Kollias et al. (2020). A compactness of cloud radars allows for their utilization on mobile platforms. A large number of cloud radars are capable of polarimetric measurements. These measurements possess a high potential that is yet to be fully exploited. For example, polarimetric cloud radars are among a few instruments for remote

sensing of particle shape in natural clouds (Matrosov et al., 2012; Myagkov et al., 2016a). Polarimetric measurements from a cloud radar confirm that during the formation phase, natural ice particles have similar shape-temperature dependencies as those observed in laboratories (Myagkov et al., 2016b). Motivated by this similarity, the German weather service (DWD) has recently introduced a habit prediction into the microphysical model McSnow (Welss et al., 2024).

Due to strong attenuation by atmospheric gases and liquid water, cloud radars have a limited spatial coverage, which is orders

of magnitude smaller than the coverage by operational centimeter-wavelength radars. During a strong rain event, cloud radar observations at, e.g., the W-band are often limited to 0.5–1 km. Despite this limitation, cloud radars provide unique information about clouds and precipitation, which considerably complements observations from other operational instruments. In addition, the ongoing project WIVERN (Illingworth et al., 2018) proposes to have the first polarimetric W-band cloud radar in space and, as a result, to have unique information about clouds and precipitation on the global scale.





Ground-based polarimetric cloud radars can provide spectral polarimetric measurements. These measurements include a set of variables similar to those measured by operational polarimetric centimeter-wavelength radars. Cloud radars can sample these variables separately for particles coexisting in a scattering volume but moving with different radial – relative to the radar – velocities. Aydin and Lure (1991) made a theoretical study simulating polarimetric spectra for 94 and 140 GHz. The simulated spectra of differential reflectivity $Z_{dr}$ show an oscillatory behavior at drop sizes roughly proportional to half of the radar

wavelength. Myagkov et al. (2020) have recently shown these oscillations in real cloud radar measurements although there has been no attempt yet to compare the exact shape of the empirical and theoretical oscillations. Aydin and Lure (1991) use fixed size-shape-velocity relations and the widely-used spheroidal approximation of the drop shape and the T-matrix scattering model (Mishchenko et al., 1996; Leinonen, 2014). In natural rain, however, the evolution of the drop properties is a stochastic process affected by numerous effects (Pruppacher and Klett, 1997, chapter 10 therein). First, the drop shape may considerably

deviate from equilibrium due to e.g. oscillations (Tokay et al., 2000; Beard et al., 2010; Szakáll et al., 2010), collision (Szakáll et al., 2014), and breakup (Villermaux and Bossa, 2009). Second, a number of studies show an evidence of sub-/super-terminal raindrops, i.e. falling with velocities considerably slower/faster than expected (Montero-Martínez et al., 2009; Thurai et al., 2013; Larsen et al., 2014). Super-terminal raindrops are likely formed by break-up of big drops (Villermaux and Eloi, 2011) while occurrence of sub-terminal drops might be related to increased drag of deformed drops (Thurai et al., 2013). Third,

turbulence is an additional factor affecting the velocity and shapes of drops (Thurai et al., 2019, 2021).

    The above-mentioned effects may cause a considerable difference between simulations and measurements at millimeter wavelengths in rain. We have identified a number of issues potentially indicating that existing size-shape-velocity approximations do not explain polarimetric observations at W-band. First, the spectra of $Z_{dr}$ simulated by Aydin and Lure (1991) oscillate around 0 dB and therefore the authors concluded that the integral $Z_{dr}$ in rain measured at W-band should not exceed

0.12 dB in rain rates up to 150 mm h$^{-1}$. Since such $Z_{dr}$ values are often in the order of measurement uncertainty, one can conclude that integral $Z_{dr}$ measurements at W-band are not informative as e.g. was done in Myagkov et al. (2020); Unal and van den Brule (2024). In real rain measurements, however, we do see $Z_{dr}$ considerably exceeding 0.12 dB. Second, Myagkov et al. (2020) suggested a self-consistency calibration evaluation which uses relations between the equivalent radar reflectivity factor at the horizontal polarization $Z_h$ (radar reflectivity hereafter) and integrated polarimetric observables. According to our

experience the approach often gives inconsistent results in case the backscattering phase $\delta$ – a proxi of median drop diameter – is below 2°. Third, a recent study from Unal and van den Brule (2024) shows a considerable discrepancy between the median diameter retrieved from cloud radar polarimetric observations and the one from a disdrometer. Interestingly, the demonstrated discrepancy is mostly pronounced at median diameters below 1.5–2 mm, while for larger median diameters the agreement is good.

The listed differences motivate the development of a better approach to simulate polarimetric variables at millimeter wavelengths. One possible way is to develop a more sophisticated stochastic model that accounts for shape disturbances and velocity deviations for each individual drop. However, this development may encounter several issues. First, it requires precise 3-dimensional shape and velocity measurements of a large number of drops, especially in natural rain, where their properties are influenced by previously discussed effects. Second, the model requires a scattering database with a vast number of perturbed





drops, necessitating computationally expensive DDA (Discrete Dipole Approximation; Chaumet, 2022) calculations. Third, a mathematical apparatus is needed to quickly calculate radar variables using the scattering properties of individual drops.

In this study, we introduce an alternative approach called the Empirical Scattering Model (ESM). This approach uses Doppler observations in rain to infer the averaged scattering properties of drops under natural conditions. For the ESM, it is necessary to select specific environmental conditions to decouple the scattering properties of drops from air movements. The main advantage

of this approach is that the inferred scattering properties inherently account for the microphysical processes of drop evolution, thereby resulting in superior accuracy compared to a model-based approach.

The main goals of this study are (1) to demonstrate that the currently known fixed size-shape-velocity relations cannot adequately explain polarimetric observations at 94 GHz in rain, (2) to suggest the ESM for $Z_{dr}$ and $\delta$, and (3) to show implications of the ESM for integral polarimetric cloud-radar observables. Section 2 introduces a cloud radar and in situ

instrumentation used throughout the study. Processing of the spectral radar data is explained in details in Sec. 3. Section 4 shows how to assign drop sizes to individual spectral lines of spectral measurements from the radar. The ESM is introduced in Sec 5. The ESM is then used in Sec.6 to explain and mitigate the issues in existing polarimetry-based techniques. Section 7 summaries the obtained results and provides an outlook.

## 2    Instrumentation and processing

This section introduces a cloud radar and several in situ rain-sampling tools utilized in the research. As all the instruments have been previously detailed in literature, we only provide specifics that are crucial for this study. More comprehensive information about the operation of these instruments can be located in the provided references.

### 2.1    W-band cloud radar

The main instrument used in this study is a W-band cloud radar manufactured by Radiometer Physics GmbH (RPG), Germany.

The radar uses an FMCW (Frequency Modulated Continuous Wave) signal and features the STSR (Simultaneous Transmission and Simultaneous Reception) polarimetric mode, also known as the hybrid mode (Bringi and Chandrasekar, 2001, Sec. 4.7 therein). Details of the radar's operation principles are given in Küchler et al. (2017). The main radar specifications are listed in Table 1. During operation the radar provides spectra of radar reflectivity $Z_h(v_k)$, differential reflectivity $Z_{dr}(v_k)$, differential phase $\Phi_{dp}(v_k)$, and correlation coefficient $\rho_{hv}(v_k)$ with $v_k = V_k/\sin\phi$. $V_k$, $\phi$, and $k$ are the radial velocity, the elevation angle,

and the index of a spectral line, respectively. Hereafter, we refer to $Z_h(v_k)$, $Z_{dr}(v_k)$, $\Phi_{dp}(v_k)$, and $\rho_{hv}(v_k)$ measured in a range bin at a certain time, as a set of spectra. As shown in Küchler et al. (2017), the radar uses several chirp types to sample an atmospheric profile. Within this study, however, we only focus on measurements collected with a chirp type used to sample the 1.2 km distance closest to the radar. This helps us to avoid analysis of measurements taken with chirps having different settings (e.g. range and Doppler resolution). In addition, at these close distances the signal-to-noise ratio is high and therefore

the random error in spectral polarimetric variables is low (Myagkov and Ori, 2022).





**Table 1.** Typical specifications of the RPG W-band cloud radar. FMCW stands for Frequency-Modulated Continuous Wave.

| Parameter | Value |
| --- | --- |
| Center frequency [GHz] | 94 |
| Transmitted power [W] | 1.5 |
| Signal type | FMCW |
| Antenna gain [dBi] | 50.1 |
| Half-power beam width [°] | 0.56 |
| Sensitivity at 5 km, 10 s sampling, and 30 m range resolution [dBZ] | −45 |
| Scanning range, azimuth [°] | 0–360 |
| Scanning range, elevation [°] | 0–180 (horizon-to-horizon) |

The radar unit has been previously used in a number of studies (e.g. Myagkov et al., 2020; Acquistapace et al., 2022; von Terzi et al., 2022). Since 2021, the radar has been used to collect spectral polarimetric observations in rain at three locations with different precipitation climatology. Details about the locations and operational modes of the radar are given in Table 2.

## 2.2 Thies disdrometer with event mode

Since the radar has Doppler capabilities, it can accurately measure the radial velocity of raindrops. In the case of non-zenith observations, the measured radial velocities reflect not only terminal velocity of drops but also contributions from air motions and horizontal wind in particular. Horizontal wind as well as up- and downdrafts shift measured Doppler spectra and, thus, the measured absolute radial velocity cannot be directly assigned to drop size. In order to constrain the absolute velocity, we use long-term observations from a Thies disdrometer (Fehlmann et al., 2020) that is permanently operated at the RPG facility in
Meckenheim, Germany. The disdrometer is operated in the event mode described in the instrument manual. In the event mode, the disdrometer provides measurements of size and velocity for each individual particle with resolution much better than the grid of the standard measurement regime. The event mode has been previously used for the calibration evaluation in Myagkov et al. (2020). In this study, continuous measurements from May 2020 to the end of June 2023 are used.

## 2.3 Supplementary in situ instruments

The radar is equipped with a weather station WTX530 from the Vaisala company. Within the study, we use surface temperature, relative humidity, and pressure from this instrument.

For evaluation purposes in Sec. 6.2, we use a Thies disdrometer operating in the standard mode. The disdrometer is an operational unit installed at the Hohenpeißenberg observatory (Location 1 in Table 2). The disdrometer was located within 10 m from the cloud radar.



**Table 2.** Locations and measurement settings used to collect spectral polarimetric observations in rain. Measurement settings are given only for the chirp type used to measure distances close to the cloud radar. [1] Due to strict data policy in China, we are advised not to provide the name of the site operator and exact coordinates for the Location 2.

| Parameter | Location 1 | Location 2 | Location 3 |
|---|---|---|---|
| Location | Hohenpeißenberg, Germany | Shanghai, China | Tenuta Cannona, Italy |
| Site operator | DWD | Chinese colleagues of RPG[1] | Agrion Foundation |
| Coordinates | 47°48'03"N 11°00'40"E | 31°14'N 121°28'E[1] | 44°40'55"N 8°37'29"E |
| Altitude above see level [m] | 968 | 4 | 287 |
| Typical surface pressure [hPa] | 900 | 1000 | 985 |
| Period | July 2021 – October 2021 | June 2022 – December 2022 | May 2023 – May 2024 |
| Range interval [m] | 100–1233 | 100–1233 | 100–1233 |
| Range resolution [m] | 29.8 | 3.7 | 7.5 |
| Nyquist velocity range [m s$^{-1}$] | ±7.33 | ±7.45 | ±7.36 |
| Doppler resolution [cm s$^{-1}$] | 2.86 | 2.88 | 5.75 |
| Integration time [s] | 3 | 1.1 | 2.2 |

## 3 Selection and processing of cloud radar spectra

The signal contribution to the radar spectra is defined by scattering from particles moving with radial velocities in the range from $v_k - \Delta v/2$ to $v_k + \Delta v/2$, where $\Delta v$ is the Doppler resolution. The radial velocities are defined not only by terminal velocities of raindrops and elevation angle, but also by air motions and scanning (Doviak et al., 1979). We exclusively use non-scanning data in this study to avoid spectral broadening due to scanning. The radar used in this study has a narrow beam and small range resolution, and therefore, the contribution of the wind shear within the sampling volume is small. Turbulence is a major contributor, leading to a mixture of raindrops with considerably different terminal velocities in a spectral line. The effect of turbulence of Doppler measurements is highly variable. Some studies show that effects of turbulence can be mitigated or even characterized in a retrieval of DSD (e.g. Moisseev and Chandrasekar, 2007; Tridon and Battaglia, 2015). However, such a retrieval relies on assumed shape approximation, scattering model, and relations between raindrop properties. In this study, however, we avoid using these assumptions whenever possible. Therefore, spectra measured under low-turbulence conditions must be selected for the following analysis. This section presents an algorithm used to identify such spectra.

### 3.1 SNR-based selection

We exclude all observations at distances smaller than 290 m. This is done to avoid any effects related to near-field and incomplete overlap between the transmitting and receiving antennas. Then, sets of spectra with $Z$ below 5 dBZ are excluded from the analysis. The threshold is empirically chosen to exclude clouds and atmospheric plankton and, on the other hand, to

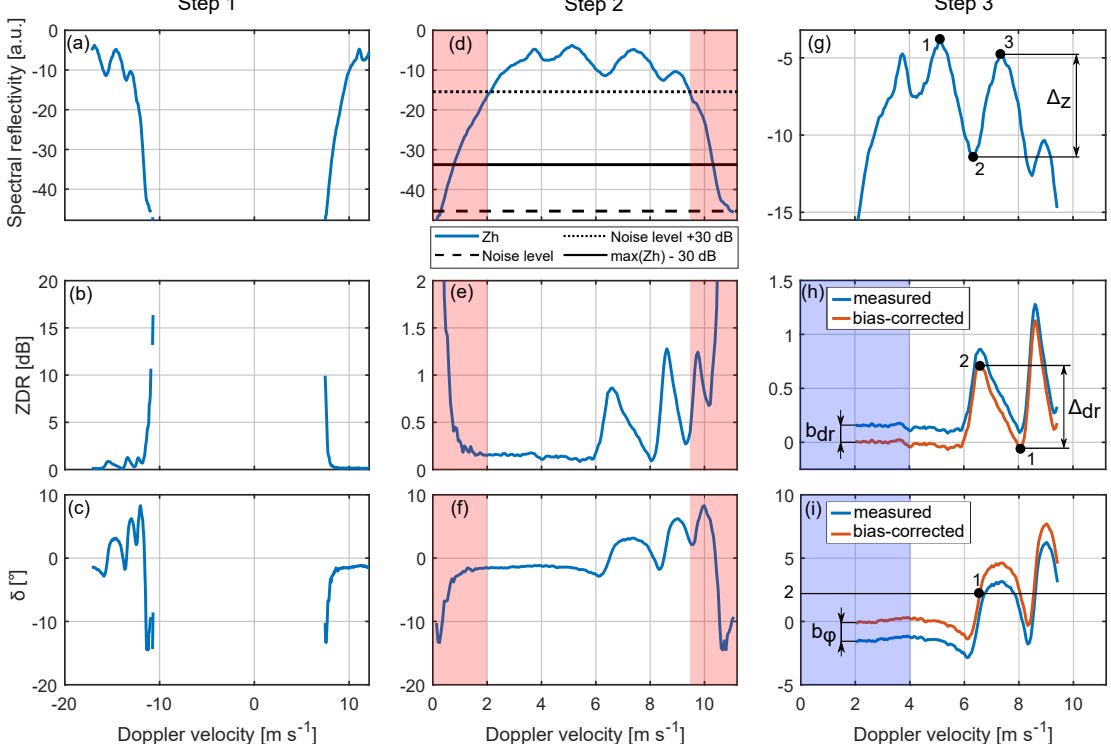

**Figure 1.** Processing of an individual spectral set. The case was arbitrary chosen from the dataset from location 1. This figure is provided only to illustrate the processing steps. Upper, middle, and lower rows correspond to spectral reflectivity, spectral differential reflectivity, and spectral backscattering differential phase shift, respectively. The left column shows spectra after the thresholding. The middle column shows alias-corrected spectra. This column also shows the selection of spectral lines based on SNR (Sec. 3.1). The red-shaded areas mark spectral lines non-fulfilling the requirements and therefore excluded from the analysis. The dashed line shows the noise level. The dotted line shows the 30 dB SNR level. The solid line shows the level 30 dB lower than the maximum spectral line. The right-most column illustrates the bias correction (Sec. 3.2) and the utilization of resonance effects as explained in Sec. 3.3. The blue solid lines in the right-most column corresponds to spectra after the SNR selection. Red lines show bias corrected backscattering polarimetric variables.

still include observations in rain affected by strong attenuation. The following steps are illustrated in Fig. 1. We check for the aliasing effect (Blackman and Tukey, 1958, Sec. B.12 therein). If a set of spectra is aliased, its left and right parts are glued to get continuous spectra. The absolute velocity is then roughly corrected by setting the left-most detected $Z_h(v_k)$ corresponding to drops with smallest fall velocity to 0 m s$^{-1}$. The velocity correction for air motions will be done in Sec. 4 therefore an

165 accurate correction of the velocity is not important at this step. Next, all lines with SNR below 30 dB are removed from a set of spectra. This is done to minimize the random measurement error in spectral polarimetric variables (Myagkov and Ori, 2022). Also, spectral lines smaller than 30 dB relative to the maximum spectral line in the analyzed spectrum are removed to exclude effects related to the spectral leakage (Harris, 1978) due to FFT (Fast Fourier Transform).



### 3.2 Correction of biases in polarimetric measurements

The spectral backscattering differential reflectivity $z_{dr}(v_k)$ and backscattering phase $\delta(v_k)$ are derived as follows:

$$z_{dr}(v_k) = Z_{dr}(v_k) - b_{dr} \tag{1}$$
$$\delta(v_k) = \Phi_{dp}(v_k) - b_{\phi}, \tag{2}$$

where $b_{dr}$ and $b_{\phi}$ are biases. These biases are estimated using the method presented in Myagkov et al. (2020, Sec. 3.3 therein) by averaging $Z_{dr}(v_k)$ and $\Phi_{dp}(v_k)$ in the range of $v_k$ corresponding to spherical raindrops. Spherical raindrops produce no

backscattering polarimetric effects and, therefore, all deviations from 0 dB and 0° in $Z_{dr}(v_k)$ and $\Phi_{dp}(v_k)$, respectively, are due to calibration (Ryzhkov et al., 2005a; Myagkov et al., 2016a; Cao et al., 2017), antenna properties (Chandrasekar and Keeler, 1993; Mudukutore et al., 1995), and propagation effects (Trömel et al., 2013; Myagkov et al., 2020). In this study the averaging is performed over $v_k$ in the range from 0 to 4 m s$^{-1}$.

### 3.3 Selection based on resonance effects

Spectra of backscattering radar observables measured in rain have a series of pronounced maxima and minima due to resonance effects (Mie scattering) at drop sizes comparable to the wavelength (Oguchi, 1983; Aydin and Lure, 1991; Kollias et al., 2007). The span between these maxima and minima can be used as a proxy of the turbulence strength. In a low-turbulence environment, the span is the highest, while in the case of turbulence, the span reduces due to spectral broadening. In order to select sets of spectra with low turbulence, we apply a series of checks. First, we exclude all sets of spectra with $\delta(v_k)$ not exceeding 2° in at

least one spectral line. Then, we identify the spectral line with the maximum $Z_h(v_k)$ (i.e. maximum spectral reflectivity, point 1 in Fig. 1g) in each set. Using the part of the $Z_h(v_k)$ spectrum with $v_k$ exceeding the one of the maximum spectral line, we find the minimum (point 2 in Fig. 1g) and the following maximum (point 3 in Fig. 1g)). The span $\Delta_Z$ between these minimum and maximum, i.e. between points 3 and 2, should exceed 6 dB. Note that the thresholds used in these conditions are empirically chosen to exclude a majority of spectra with turbulence. The strongest condition is to exclude all sets of spectra in which the

span $\Delta_{dr}$ between the first maximum and the following minimum in $z_{dr}(v_k)$ (points 2 and 1 in Fig. 1h, respectively) does not exceed 1 dB.

Finally, in each set of spectra fulfilling the above-mentioned criteria, we search for the first spectral line $\delta(v_k)$ exceeding 2° (point 1 in Fig. 1i). All sets of spectra are adjusted so that this spectral line corresponds to the same velocity. The absolute value of this velocity is arbitrarily chosen between 6 and 7 m s$^{-1}$ and is not important at this stage.

### 195 3.4 $\rho_{hv}$-based quality check of selected spectra

In total 2127, 3407, and 9021 spectra from the locations 1, 2, and 3 satisfy all the above-mentioned conditions, respectively. For illustration purposes Fig. 2 displays the statistics of the spectra selected for the location 3. The variability in $Z_h(v_k)$ (Fig. 2a) is mostly defined by the DSD variability. The range of $z_{dr}(v_k)$ and $\delta(v_k)$ is mostly within $\pm0.1$ dB and $\pm0.1°$ relative to corresponding median values, respectively. This low variability indicates that the polarimetric variables are nearly the same in



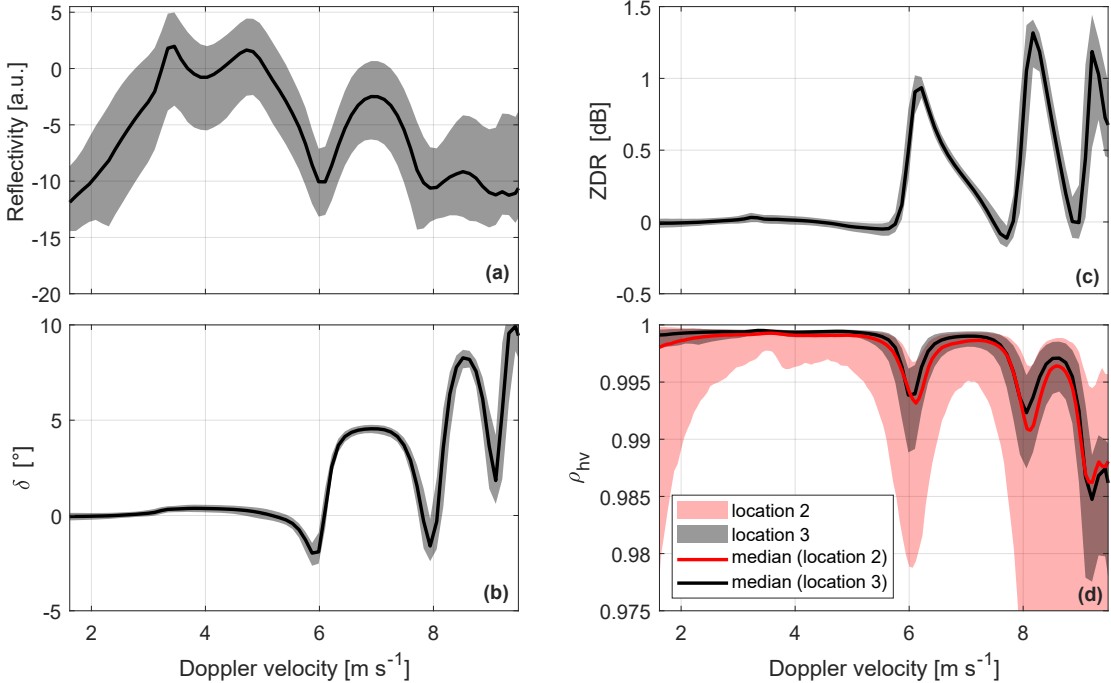

**Figure 2.** Statistics of the selected spectral sets for the location 3. Spectral reflectivity (a), backscattering phase (b), differential reflectivity (c), and correlation coefficient (d) are shown. Solid black lines and black shaded areas correspond to mean values, and 5 and 95 percentiles, respectively. The red solid line and the red-shaded area correspond to mean values, and 5 and 95 percentiles of the spectral correlation coefficient calculated for the location 2 before the additional selection criterion described in Sec.3.4.

all selected spectral sets. In order to compare quality of the selected spectra among the sites, we additionally checked statistics of $\rho_{hv}(v_k)$, since this parameter is sensitive to enhanced measurement error, mixture of particles with different scattering properties, and increased variability in the canting angle. Figure 2d shows that selected spectra of $\rho_{hv}(v_k)$ at location 2 are on average considerably lower than those at location 3. Since the same radar unit was used at all 3 locations, the difference is not likely caused by the antenna system (Mudukutore et al., 1995). Therefore, the lower values of $\rho_{hv}(v_k)$ can be caused

by one or a combination of the following factors: (1) smaller resolution volume and shorter integration time leading to higher measurement errors, (2) stronger effect of turbulence leading to broader distribution of drop sizes and orientation angles in each spectral line, and (3) scattering from large individual drops in a resolution volume down to 25 m$^3$ might not be volume distributed (Schmidt et al., 2012, 2019). Exact reasons for the lower $\rho_{hv}(v_k)$ seen at the location 2 are out of the scope of this study. For the following analysis, we introduce an additional rule only for the location 2. As indicated in Fig. 2, $\rho_{hv}(v_k)$ for the

location 2 and exceeding corresponding median values are similar in magnitude to $\rho_{hv}(v_k)$ observed at location 3. Therefore, within a spectral set, we exclude all spectral lines with $\rho_{hv}(v_k)$ below corresponding median values.

Finally, for each location we use the selected spectra to find median spectra in order to reduce spectrum-to-spectrum variability in polarimetric variables. The median variables are further denoted as $\overline{Z_h}(v_k)$, $\overline{z_{dr}}(v_k)$, and $\overline{\delta}(v_k)$. Note, that since $\rho_{hv}(v_k)$





is prone to statistically significant biases related to radar-specific characteristics such as noise level and antenna quality, $\rho_{hv}(v_k)$
is not further analyzed in this study.

## 4 Assignment of drop size

In the previous section we selected sets of spectra measured under low-turbulence conditions. These sets include natural variability of drop properties. There are, however, two problems. First, the terminal velocity of raindrops has to be assigned to spectral lines. Second, the size-velocity relations need to be derived to assign drop size to each spectral line. This section
attempts to solve these issues.

### 4.1 Velocity parameterization

We introduce the following parameterization of relations between an equivolumetric drop diameter $D$ and the velocity $V$ corresponding to a spectral line:

$$V(D) = \overbrace{\underbrace{\left[\tanh(a_1 D + a_2) - \tanh(a_2)\right] a_3 \left(\frac{\rho_0}{\rho_a}\right)^{0.4}}_{V_t} \sin(\phi)}^{V_0} + V_b, \tag{3}$$

where coefficients $a_{1,2,3}$ are fitting parameters, tanh is the hyperbolic tangential, $\rho_0$ and $\rho_a$ are air densities at 1000 hPa and at the measurement site, respectively, $\phi$ is an elevation angle (30° throughout this study), $V_b$ is a bias in velocity due to e.g. wind, $V_0$ and $V_t$ are the terminal velocities of the drop at atmospheric pressures of $\rho_0$ and $\rho_a$, respectively. The bias $V_b$ is the same for all spectral lines within a set of spectra. The selected function describes, with a small number of parameters, a monotonic function with adjustable curvature and with the 0 m s$^{-1}$ terminal velocity corresponding to the 0 mm diameter.
Estimation of $a_{1,2,3}$ and $V_o$ requires a certain a priori knowledge, since the measured polarimetric spectra alone are not enough to constrain the size-velocity relations. We use two additional sources of information described below.

### 4.2 Disdrometer data

The multi-year dataset from the Thies disdrometer introduced in Sec. 2.2 contains measurements of the horizontal axis $D_h$ of raindrops and the time $\tau$ required for the droplets to cross the laser beam:

$$235 \quad \tau = \frac{D_h \xi + \Delta_L}{V_0} = \frac{D \xi^{2/3} + \Delta_L}{V_0}, \tag{4}$$

where $\xi$ is the axis ratio of the raindrop ($\xi < 1$ for oblate particles) and $\Delta_L = 0.75$ mm is the laser thickness. The disdrometer data, thus, are an additional, although not self-sufficient, constrain for the size-velocity relations. The statistics of the disdrometer dataset is shown in Fig. 3. For the following analysis we select the diameter range from 0.8 to 1.2 mm to have shapes as close as possible to spherical and, on the other hand, to avoid unnatural relations between size and velocity of drops. Smaller
droplets often have unexpected velocities due to factors such as splashing, as communicated in Angulo-Martínez et al. (2018).





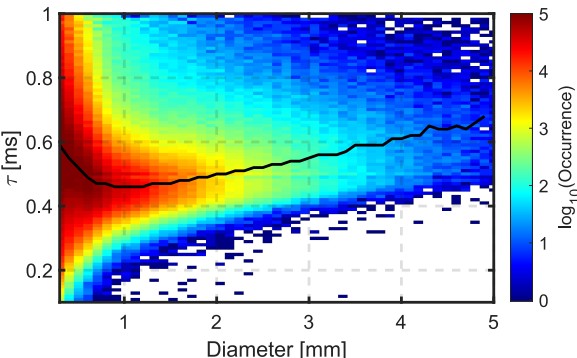

**Figure 3.** Occurrence of $\tau$ for different sizes of raindrops. The Thies disdrometer with the event mode (see Sec. 2.2) was used to collect the measurements. Measurements were taken at Meckenheim, Germany from May 2020 to the end of June 2023. The black solid line indicates $\tau_m$ for each drop diameter.

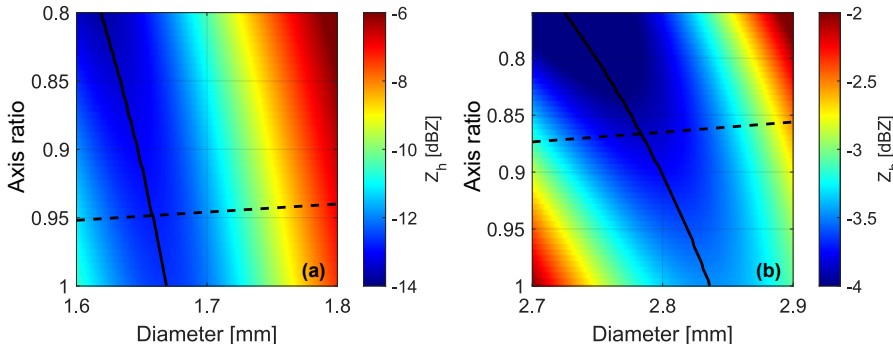

**Figure 4.** Reflectivity of a single raindrop per unit volume for different sizes and aspect ratios. The reflectivity was calculated using the SSA and the T-matrix scattering model. The vertical symmetry axis orientation was assumed. The horizontal polarization is considered. The black solid line shows the diameter at which for a given aspect ratio the reflectivity has a minimum. The black dashed line shows indicates the size-shape relation from Pruppacher and Pitter (1971). The panels (a) and (b) depict the vicinity of the first and second Mie notches, respectively.

Taking into account that raindrops smaller than 1.2 mm have near spherical shape (Thurai and Bringi, 2005, and references therein), Eq. 4 relates the terminal velocity to the disdrometer observables for these drops, assuming $\xi = 1$. Figure 3 shows that median $\tau$ (further denoted as $\tau_m$) is 0.460±0.02 ms for diameters of 0.8–1.2 mm at around 1000 hPa atmospheric pressure.

### 4.3 Simulated scattering properties

Backscattering properties of raindrops are often simulated using the T-matrix model assuming the spheroidal shape approximation (SSA hereafter). In contrast to a majority of studies with fixed size-shape relations, we calculate the backscattering





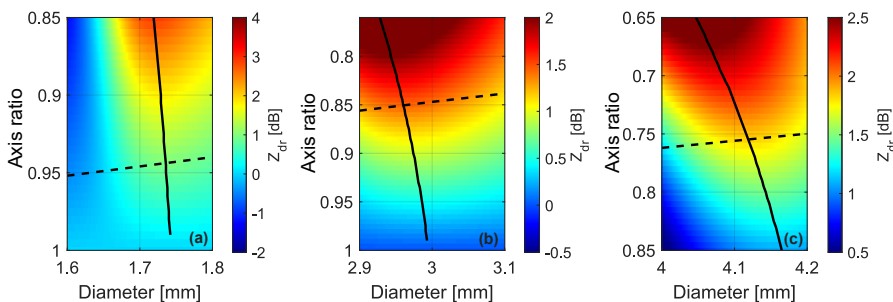

**Figure 5.** Differential reflectivity of raindrops for different sizes and aspect ratios. The differential reflectivity was calculated using the SSA and the T-matrix scattering model. The vertical symmetry axis orientation was assumed. The horizontal polarization is considered. The black solid line shows the diameter at which for a given aspect ratio the differential reflectivity has a maximum. The black dashed line shows indicates the size-shape relation from Pruppacher and Pitter (1971). The panels (a), (b), and (c) depict the vicinity of the first three maxima in differential reflectivity, respectively.

properties for different combinations of drop size and shape. Figures 4 and 5 show the results of the calculations for $Z_h$ and $z_{dr}$, respectively, in the vicinity of the resonance drop sizes. The calculations indicate that the minima in $Z_h$ and maxima in $z_{dr}$ have a relatively low sensitivity to $\xi$. The first two minima in modeled $Z_h$ correspond to equivolumetric diameters $D_1 = 1.66\pm0.02$ and $D_2 = 2.79\pm0.04$ mm, respectively. The first three maxima in modeled $z_{dr}$ occur at $D_3 = 1.73\pm0.01$, $D_4 = 2.96\pm0.02$, and $D_5 = 4.13\pm0.04$ mm. These findings will be further used to assign drop sizes to specific spectral lines of low-turbulence spectra.

### 4.4 Estimation of $a_{1,2,3}$ based on the radar and disdrometer

In order to estimate $a_{1,2,3}$, a variational approach is used. We specify a state vector $\boldsymbol{x}_v$:

$$\boldsymbol{x}_v = \begin{bmatrix} a_1 & a_2 & a_3 & V_b \end{bmatrix}^T, \tag{5}$$

where $T$ denotes the transposition. For a given $\boldsymbol{x}_v$, equivolumetric diameters can be assigned to all spectral lines using an inversion of Eq. 3. The assigned diameters are further denoted as $\hat{D}$. After the assignment, we search for equivolumetric diameters $\hat{D}_1$ and $\hat{D}_2$ corresponding to first two Mie minima in $\overline{Z_h}(v_k)$ and $\hat{D}_3$, $\hat{D}_4$, and $\hat{D}_5$ corresponding to first 3 maxima in $\overline{Z_{dr}}(v_k)$.

After the assignment, a cost function is calculated:

$$C_v = \boldsymbol{e}_v^T \boldsymbol{\Sigma}_v^{-1} \boldsymbol{e}_v. \tag{6}$$

In Eq. 6 $\boldsymbol{e}_v$ is a vector of errors based on findings from Sec. 4.2 and 4.3:

$$\boldsymbol{e}_v = \begin{bmatrix} e_1 & ... & e_6 \end{bmatrix}^T, \tag{7}$$





**Table 3.** Ranges of the state vector elements and derived solutions for locations 1–3.

| Element | Minimum | Maximum | Location 1 | Location 2 | Location 3 |
|---|---|---|---|---|---|
| $a_1$ | 0 | 1 | 0.238 | 0.243 | 0.246 |
| $a_2$ | -3 | 3 | 1.876 | 2.047 | 2.144 |
| $a_3$ | 0 | 500 | 223.376 | 306.679 | 370.382 |
| $V_b$ [m s$^{-1}$] | -2 | 2 | –0.260 | –0.633 | –0.451 |

where $e_{1..5} = D_{1..5} - \hat{D}_{1..5}$ and

$$e_6 = \sqrt{\sum_{j=1}^{N_L} (\tau_j - \tau_m)^2}. \tag{8}$$

$e_6$ is calculated using $N_L$ spectral lines corresponding to $0.8 < D < 1.2$ mm.

Diagonal elements of $\mathbf{\Sigma}_v$ are variances $\sigma_{1..6}^2$ of corresponding errors. Following Sec. 4.2 and 4.3 we set $\sigma_{1..5}$ to 20, 40, 10, 20, and 40 $\mu$m, respectively. $\sigma_6$ is set to 20 $\mu$s. Off-diagonal elements of $\mathbf{\Sigma}_v$ are set to 0 assuming no correlation between the errors.

The elements of the state vector $\boldsymbol{x}_v$ are perturbed to minimize the cost function $C_v$. We use the differential evolution algorithm (DE, Das et al., 2009). Its implementation is available in the "optim" package of Octave. In general any other minimization approach can be used. DE is a stochastic algorithm used to search for a global minimum. We employ the standard DEGL/SAW/bin strategy. We set the mutation factor to 0.8 and the crossover probability to 0.9. We also establish a tolerance level of $10^{-3}$. The process runs maximum of 100 iterations. The population size is set to 1000 times the number of elements in the state vector. The Differential Evolution (DE) algorithm halts either when it hits the maximum iteration count or when the relative difference in the cost function between the best and worst state vectors in the population falls below the set tolerance. Upon reaching a stopping criterion, the state vector that yields the lowest cost function is selected as the final output. DE does not require a priori $\boldsymbol{x}_v$ and Jacobian. Instead, it requires boundaries for the elements of the state vector. Boundaries for the state-vector parameters are given in Table 3.

The estimation procedure is separately applied to the 3 locations. Solutions for each site are given in Table 3. Figure 6 illustrates the estimated size-velocity relations. These relations agree well with the reference (Thurai and Bringi, 2005, Eq. 14 therein) and each other (Fig. 6b). The estimated terminal velocities are slightly lower than the reference one for drop sizes from 1 to 4 mm. The difference does not exceed 5%. For a majority of applications this difference is negligible. For retrievals based on polarimetric spectra, however, this difference is considerable since it affects the difference in velocity between spectral lines measured with an accuracy of a few cm s$^{-1}$. The derived size-velocity relations (Eq. 3 and Table. 3) are used to map $\overline{Z_h}(v_k)$, $\overline{z_{dr}}(v_k)$, and $\overline{\delta}(v_k)$ into the drop-size space, i.e. to obtain $\overline{Z_h}(D)$, $\overline{z_{dr}}(D)$, and $\overline{\delta}(D)$.



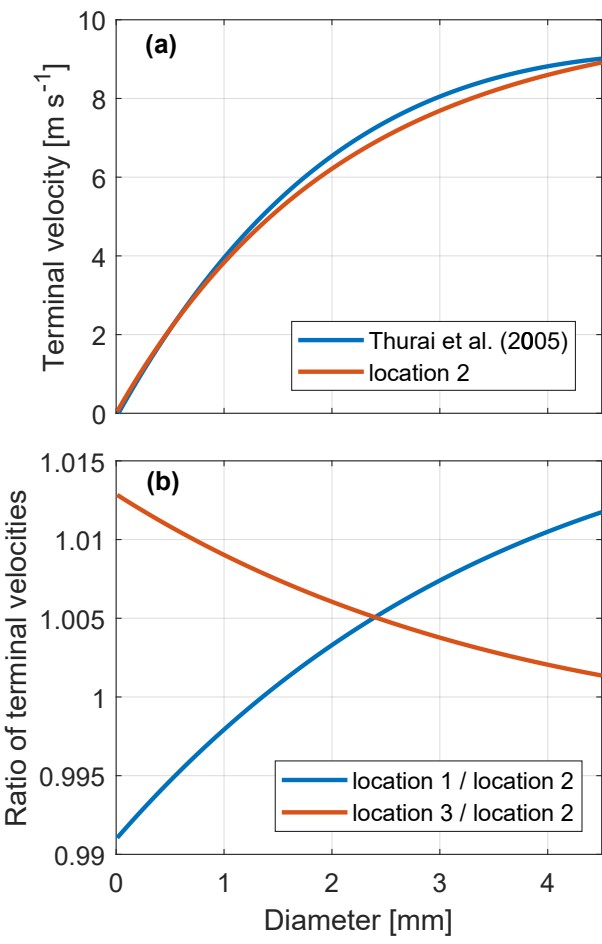

**Figure 6.** Absolute (a) and relative (b) terminal velocities as functions of drop size. The blue solid line in panel (a) corresponds to Eq. 14 in Thurai and Bringi (2005). The red solid line in panel (a) corresponds to the size-velocity relation derived for the location 2. The blue and red solid lines in panel (b) show the derived size-velocity relations derived for the locations 1 and 3, respectively, relative to the derived relation for the location 2. The terminal velocities are for the 1000 hPa pressure.

## 5 Empirical model for backscattering polarimetric variables

In the previous sections we selected spectra with minimum effect of turbulence. Using these spectra and the supplementary information we derived size-velocity relations, which allow us to assign radar variables to different drop sizes. In this section, we provide a discussion of the derived dependencies and propose an ESM to be used instead of the SSA.


## 5.1 Reflectivity spectra

Figure 7 shows dependencies $\overline{Z_h}(D)$, i.e. of the reflectivity on drop size. Note, that in this study, we aim to retrieve neither quantitative parameters of DSDs nor backscattering cross sections of individual raindrops. The main goal of this subsection is to show common effects observed in the reflectivity spectra with minimum turbulence effects.

At all the locations reflectivity peaks in the 0.6–1.3 mm diameter range. At larger sizes, the reflectivity decays with alternating maxima and minima due to resonance effects. The resonance notches are located at specific diameters roughly proportional to the radar half-wavelength. The magnitude difference between the locations is due to different DSD properties resulting in different scattering and attenuation. These differences are not important for the current study and are out of the scope.

    The blue solid line in Fig. 7 shows the reflectivity of a single drop per unit volume as a function of the drop size. This

reflectivity is derived using the SSA, size-shape relation from Pruppacher and Pitter (1971), and the T-matrix scattering model. The position of the resonance notches agrees well with those derived from the measurements. This is expected since the position of the notches is used in the optimization algorithm (Sec. 4.4). For drops larger than about 2 mm in diameter, the single drop reflectivity exceeds the median measured reflectivity because of smaller concentration of these drops and attenuation by liquid and gas.

In general, the spectral reflectivity observations agree well with previous spectral observations at W-band in rain (e.g. Kollias et al., 2007; Tridon and Battaglia, 2015). We, however, would like to draw the reader's attention to an interesting phenomenon. In low-turbulence spectra at all locations we observe a pronounced maximum at around 0.7 mm diameter. These drops have about 3.5 m s$^{-1}$ terminal velocity and do not produce any noticeable polarimetric signatures (Fig. 2). The maximum is not predicted by the SSA (blue solid line in Fig. 7). We see two possible explanations for the maximum. One hypothesis is that

this is due to scattering resonance caused by drop shapes diverging from the ideal spheroid. Another possible explanation is an increased concentration of drops with 0.7 mm due to a specific formation process (e.g break up). The exact reason of this phenomenon requires further laboratory and in situ-based investigations and is therefore out of the scope of the current study.

## 5.2 Polarimetric spectra

Figures 8a and b show one of the main results of the current study – dependencies of $\overline{z_{dr}}$ and $\overline{\delta}$ on drop size, respectively.

The results derived independently for all the locations agree well. There are some differences visible at diameters exceeding 3 mm. These differences can be due to different range resolution, sampling time, and atmospheric pressure. The latter may have an effect because at lower atmospheric pressure the slope of the size-velocity relation is less steep, which results in a better resolution of the large-diameter drops. In the following, we select the location 1 as a reference, because observations at this location have the most coarse range resolution, the largest integration time, and the lowest pressure among the 3 analyzed

datasets. This selection is also supported by the largest spans between maxima and minima in $\overline{z_{dr}}(D)$ and $\overline{\delta}(D)$.

    Dependencies $\overline{z_{dr}}(D)$ and $\overline{\delta}(D)$ are compared with differential reflectivity $z_{dr,s}$ and backscattering phase $\delta_s$ simulated using known size-shape relations and the SSA. We use 6 relations from Thurai and Bringi (2005, Eqs. 2–7 therein) and the relation from Pruppacher and Pitter (1971). Figures 8c and d show that all simulated $z_{dr,s}(D)$ and $\delta_s(D)$ significantly deviate





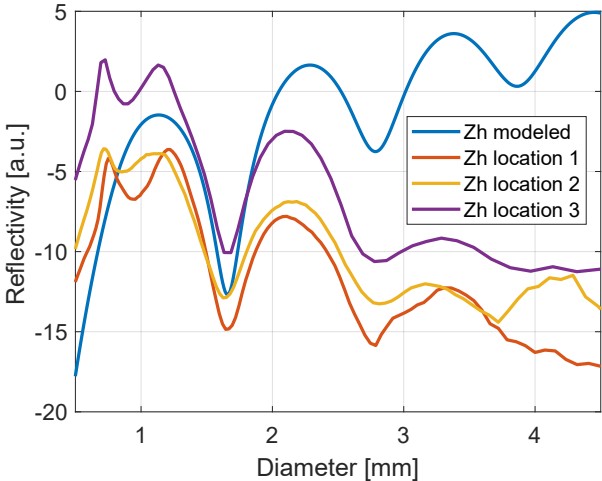

**Figure 7.** Spectral reflectivity as a function of drop diameter. The red, yellow, and purple lines correspond to median dependencies derived empirically as described in Sec. 4 from selected low-turbulence spectra at locations 1, 2, and 3, respectively. The blue solid line shows a simulated reflectivity of a single drop per unit volume. The simulation was made using the SSA and the T-matrix scattering model. The size-shape relation from Pruppacher and Pitter (1971) was assumed.

from radar observations. The maximum deviation reaches 1 dB and 7° in differential reflectivity and backscattering phase,
respectively, in the 3–4 mm-size range.

We also check how well simulated Doppler spectra fit the observed polarimetric variables. For this we use the 7 above-mentioned size-shape relations and the Eq. 14 from Thurai and Bringi (2005) for the size-velocity relation. The comparison is given in Fig. 9. In general, the simulated variables explain the "oscillations" in the spectra, however their exact shape, i.e. positions and magnitudes of maxima and minima, does not fit the observations. Interestingly, magnitudes of the simulated
maxima fit the observed ones fairly well. Magnitudes of minima, in contrast, diverge considerably. Effects of turbulence and orientation cannot explain the differences. These effects reduce magnitudes of both maxima and minima. We, thus, conclude that the known size-shape-velocity relations and the SSA cannot be used to properly simulate spectral polarimetric variables at W-band.

### 5.3 Approximation of scattering properties with an artificial neural network

One solution to the above-stated problem of polarimetric spectrum simulation is to develop a more sophisticated scattering model, e.g. taking into account possible shape disturbances and oscillations. Development of such a model is a challenging task and is out of the scope of the current study. We propose an easier solution – the ESM. The dependencies $\overline{z_{dr}}(D)$ and $\overline{\delta}(D)$ can be approximated with a function and this approximation can be then used for simulation of polarimetric observables. In meteorological studies a polynomial approximations are often used. However, taking into account the complex oscillatory
behavior of the functions and a high requirement for the fitting accuracy, we decided to use an artificial neural network for the





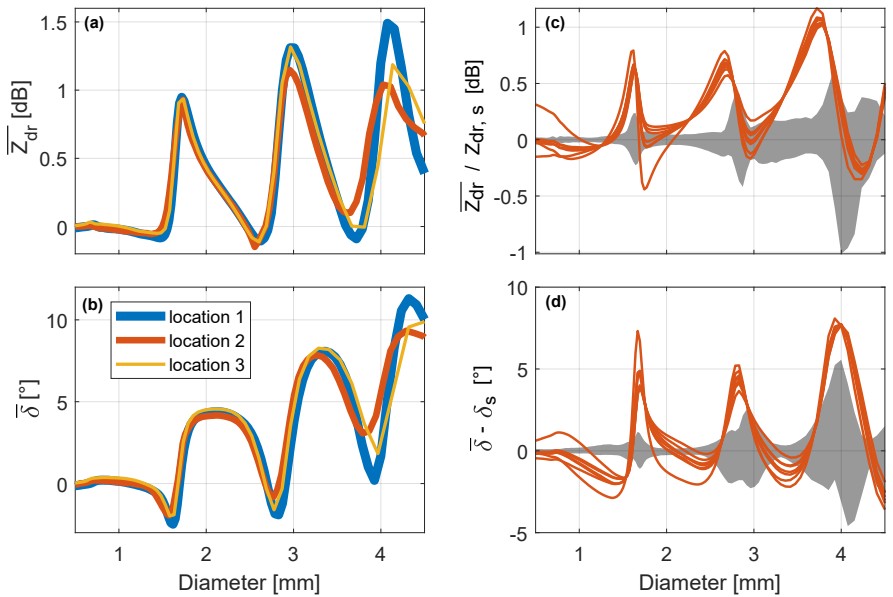

**Figure 8.** Spectral differential reflectivity (a) and backscattering phase (b) as functions of a drop diameter. The blue, red, and yellow lines in (a) and (b) correspond to median dependencies derived empirically as described in Sec. 4 from selected low-turbulence spectra at locations 1, 2, and 3, respectively. Red lines in panel (c) depicts ratios of the experimental size spectrum of the differential reflecticity from the location 1 over simulated ones. The gray shaded area in (c) illustrates the spectrum-to-spectrum variability of the measured spectrum. The upper and lower boundaries of the area are ratios of the median values over 5th and 95th percentiles, respectively. Red lines in panel (d) shows differences between simulated size spectra of the backscattering phase and the experimental one from the location 1. The gray shaded area in (d) illustrates the spectrum-to-spectrum variability of the measured spectrum. The upper and lower boundaries of the area are differences between the median values and 5th and 95th percentiles, respectively. Simulated spectra are derived using the SSA and the T-matrix scattering model. 7 different size-shape relations were used, 6 from Thurai and Bringi (2005, Eqs. 2–7 therein) and the one from (Pruppacher and Pitter, 1971).

approximation (ANN). ANNs have become a widely used tool. We, therefore, do not aim to give basics of ANN architecture and training in this study. This information can be found in a handbook (e.g. Demuth et al., 2014). The trained ANN is provided as a supplement to this study and can be used even by a reader not familiar with ANNs.

We train ANN which takes a min-max normalized equivolumetric diameter $D_n$ as an input and outputs a vector $\boldsymbol{y}$:

$$\boldsymbol{y} = \mathbf{K}\left[\tanh\left(\boldsymbol{k}D_n + \boldsymbol{b}_1\right)\right] + \boldsymbol{b}_2, \tag{9}$$

where $\boldsymbol{k}$ and $\boldsymbol{b}_1$ are $N_1 \times 1$ vectors with weighting coefficients and biases for the hidden layer, respectively, with $N_1$ being the number of neurons in the hidden layer; $\mathbf{K}$ and $\boldsymbol{b}_2$ are a $N_2 \times N_1$ weighting-coefficient matrix and a $N_2 \times 1$ bias vector for the output layer, respectively, with $N_2$ being the number of elements in $\boldsymbol{y}$. Elements of the vector $\boldsymbol{y}$ correspond to min-max normalized output parameters. Output parameters include the reflectivity of one drop per unit volume $z_h$, differential



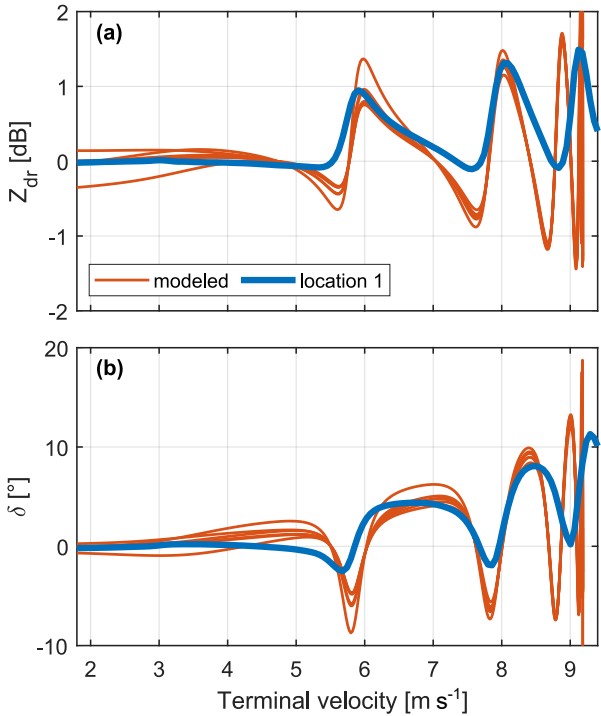

**Figure 9.** Spectral differential reflectivity (a) and backscattering phase (b) as functions of the terminal velocity at 1000 hPa. The blue solid lines correspond to median dependencies derived empirically as described in Sec. 4 from selected low-turbulence spectra at the location 1. Red lines depict corresponding simulated spectra. Simulated spectra are derived using the SSA and the T-matrix scattering model. 7 different size-shape relations were used, 6 from Thurai and Bringi (2005, Eqs. 2–7 therein) and the one from (Pruppacher and Pitter, 1971).

reflectivity $z_{dr}$, backscattering phase $\delta$, one-way attenuation $A_h$, differential attenuation $A_{dp}$, and specific phase shift $K_{dp}$. Unlike $z_{dr}$ and $\delta$, it is not possible to deduct $z_h$, $A_h$, $A_{dp}$, and $K_{dp}$ from radar observations alone. We therefore use T-matrix simulations as the best available knowledge for these variables. For the simulations of these 4 variables we assume the SSA, the size-shape from Pruppacher and Pitter (1971), and the same elevation angle as used in the observations.

The main advantage of the provided model is the better representativeness of the differential reflectivity and backscattering

phase. In addition, the calculations are quicker than running the T-matrix calculations. On the other hand there are a number of limitations. The model is only valid for drops smaller than 5 mm. This limitation, however, is often tolerable since larger drops have small concentration and do not strongly affect integrated rain characteristics. In order to get the model for different radar frequencies or elevation angles, few-month datasets with appropriate observations in rain are required. Taking into account the advantages and disadvantages, the proposed model is mainly applicable to tasks requiring a superior accuracy of backscattering

polarimetric variables at millimeter wavelengths.





## 6 Implication for integral polarimetric variables

In the previous section we presented the ESM. The main difference to existing simulation-based scattering models is that backscattering polarimetric variables are approximated from observations in rain under low-turbulence conditions. The previous section also shows that the ESM differs considerably from simulations based on existing size-shape-velocity relations for

raindrops. In this section we show how these differences affect integrated backscattering polarimetric variables in rain.

### 6.1 Integrated differential reflectivity and backscattering differential phase

In order to simulate integrated radar observables an assumption on DSD is required. A widely-used parameterization of DSD in rain is the normalized-gamma DSD given in Illingworth and Blackman (2002, Eq. 13 therein). The parameterization has a mono-modal shape and requires three variables, namely the concentration parameter $N_L$, shape parameter $\mu$, and median

diameter $D_0$. The radar observables can be calculated using the normalized-gamma DSD as explained in Appendix B.

Differential reflectivity and backscattering differential phase do not depend on $N_L$ and are often used as a proxy of $D_0$ in weather and cloud radars (Trömel et al., 2013; Bringi and Chandrasekar, 2001, Sec. 7.1 therein). We therefore fix $N_L$ at an arbitrary chosen value of 2500 m$^{-3}$ mm$^{-1}$ and calculate $z_{dr}$ and $\delta$ for $\mu$ from 0 to 15 and $D_0$ from 0.5 to 2.5 mm. The calculations are performed using the SSA and ESM as explained in Appendix B.

Figure 10 shows the results of the calculations. Two considerable differences between the two models are clearly visible. First, for $D_0$ below 1.5 mm the ESM indicates that $\delta$ is barely sensitive to $D_0$ (Fig. 10a), although a high sensitivity of $\delta$ to $D_0$ is expected from the SSA (Fig. 10c). This difference explains recent findings of Unal and van den Brule (2024), who developed a retrieval of $D_0$ based on $\delta$. Their evaluation of the retrieval showed that the derived $D_0$ is consistent with disdrometer observations for $D_0 > 1.5$ mm. For smaller $D_0$, the retrieval considerably underestimates $D_0$. The ESM shows that

for $D_0 < 1.5$ mm $\delta$ does not exceed 0.5 ° which is often close to the measurement uncertainty of this polarimetric parameter. The threshold of 1.5 mm in $D_0$ likely results from pronounced polarimetric signatures of drops with size exceeding this value (see Fig. 8). Second, the ESM disproves conclusions based on SSA (Aydin and Lure, 1991; Myagkov et al., 2020; Unal and van den Brule, 2024, also Fig. 10d) that $z_{dr}$ is not informative in rain. This study demonstrates that $z_{dr}$ is sensitive to $D_0 > 1.5$ mm and can exceed the limits expected from the SSA. For instance (Aydin and Lure, 1991) concluded that $z_{dr}$ at

94 GHz should not exceed 0.12 dB at rain rates up to 120 mm h$^{-1}$. The ESM, however, provides an evidence that $z_{dr}$ can exceed even higher values at lower rain rates. The last conclusion is an evidence that the differences reported in Sec. 5.2 cannot result from spectral broadening by turbulence, because the broadening does not affect integrated polarimetric measurements.

### 6.2 Effects on self-consistency of cloud radar measurements

Section. 6.1 shows that $\delta$ that is expected from the SSA differs considerably from true values observed in rain especially at

390 $D_0 < 1.5$ mm. It has been also shown that this difference has a considerable effect on accuracy of polarimetry-based methods. Myagkov et al. (2020) introduced a self-consistency check of the W-band reflectivity based on redundancy of information contained in radar observables at the W-band. The self-consistency approach uses $\delta$ as a proxy of $D_0$. Since the method is





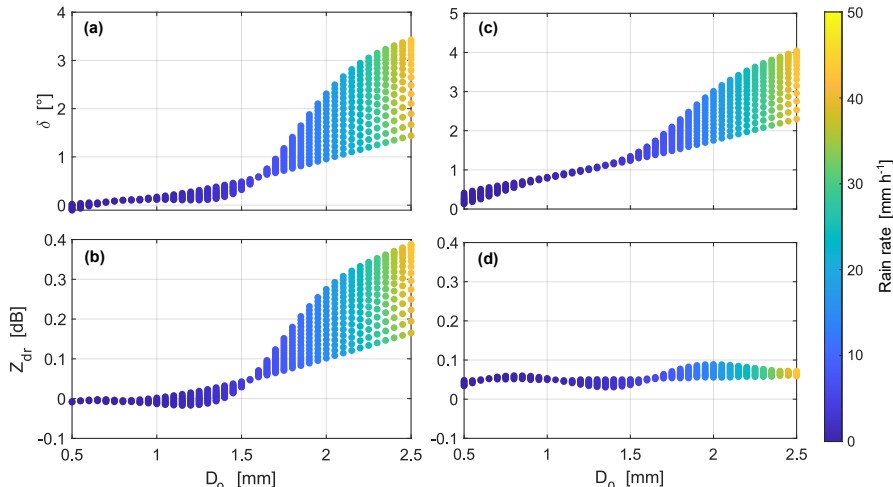

**Figure 10.** Panels (a) and (c) show dependencies between the integrated backscattering phase and the median diameter. Panels (b) and (d) illustrate relations between the integrated differential reflectivity and the median diameter. The dependencies in the first and the second columns were derived using the ESM and the SSA, respectively. The SSA assumes the size-shape relation from Pruppacher and Pitter (1971).

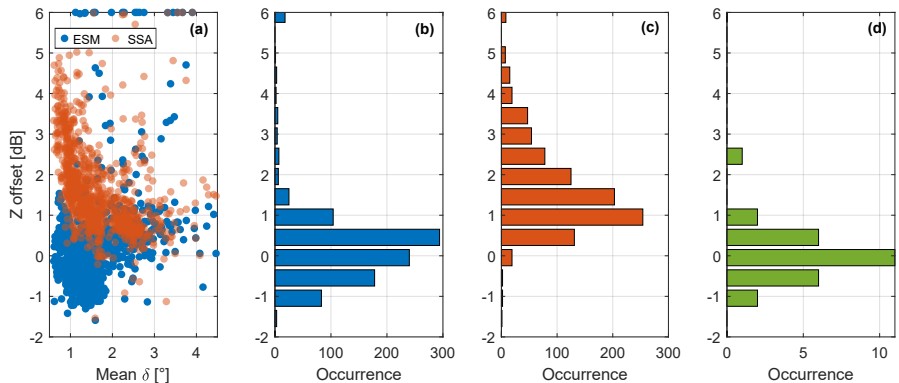

**Figure 11.** Reflectivity offsets derived using different evaluation approaches. The data from 28 rain events observed at location 1 were used. Panels (a)–(c) show results of the self-consistency method. 975 profiles were used for this method. The results in (c) and red dots in (a) correspond to the self-consistency method with the SSA while the results in (b) and blue dots in (a) were derived using the self-consistency method with the ESM. The panel (d) shows results of the disdrometer-based evaluation. Each dot in (a) and each unit in (b) and (c) corresponds to an offset derived from a single profile. A unit in (d) results from a rain event with at least 2 h duration.

based on the SSA, it can be also affected by the differences between simulated and observed $\delta$. In order to evaluate this, we applied several reflectivity checks to 28 rain events observed at location 1. First, the original SSA-based consistency method was applied (red dots in Fig. 11a and histogram in Fig. 11c). In total there are 975 profiles where this method is applicable. Second, we utilized the self-consistency method based on the ESM (blue dots in Fig. 11a and histogram in Fig. 11b). Here,





the same 975 profiles were used as for the SSA-based method. Last, as an independent evaluation method, we applied the disdrometer-based algorithm (Fig. 11d) presented in Myagkov et al. (2020). The disdrometer-based approach shows that the radar-reflectivity offset estimated from a single rain event is $-0.1 \pm 0.7$ dB (mean and standard deviation). This indicates that the radar is well-calibrated. We would like to highlight that for the disdometer-method, one offset value is derived for a rain event of at least 2-hours duration. The self-consistency method provides an estimate from a single sample with a duration in the order of few seconds. The results of the original self-consistency method show on average a 1.4 dB offset with the single-sample standard deviation of 1.3 dB. When $\delta$ exceeds $2°$ the reflectivity offset is mostly below 1 dB. At lower values of $\delta$, the estimated offsets increase up to 5 dB. This increase is not consistent with the disdromter-based results and therefore it may result from the difference between SSA and ESM. In contrast, the self-consistency method based on the ESM shows a narrower and less biased distribution of estimated offsets ($0.2 \pm 1$ dB). The distribution (Fig. 11b) is consistent with the one from the disdrometer-based method (Fig. 11d). Thus, we conclude that the ESM should be used for the self-consistency reflectivity evaluation in the case of W-band radars.

## 7 Summary and outlook

Understanding the relationships between raindrop properties – such as size, shape, and velocity – is crucial in various fields including meteorology, hydrology, agriculture, remote sensing, and telecommunication. While existing approximations for the relationships are accurate enough and have been widely used for decades, this study demonstrates that the approximations are not enough to interpret polarimetric radar observations at W-band. A possible solution for this issue is to develop a highly sophisticated model accounting for a fine geometry of drops in natural rain. Such a model, however, requires enormous efforts from the community. Instead, a much simpler approach – the empirical scattering model – is introduced in this study. The model is based on an analysis of high-quality low-turbulence Doppler spectra collected during past 3 years at different locations. These spectra allow for estimation of size-velocity relations and average backscattering polarimetric response of drops with different sizes. The main advantages of this model are high accuracy and accounting for natural microphysical effects in rain. The main disadvantage is that this model cannot be simply extended to arbitrary frequencies and elevation angles. Adaptation of the model to a given frequency and elevation angle in general requires a collection of a dataset based on which the low-turbulence spectra can be identified.

This study also demonstrates the implication of the empirical scattering model on integral polarimetric variables. In contrast to theoretical studies concluding that $\delta$ at the W-band is a proxy of the median drop diameter, we found that $\delta$ is barely sensitive to $D_0$, when $D_0$ is smaller than 1.5 mm. This finding explains the underestimation of $D_0$ in Unal and van den Brule (2024). We also disproved the conclusion (Aydin and Lure, 1991), based on available knowledge on relation between drop properties, that $z_{dr}$ is not informative at W-band. Similarly to $\delta$, $z_{dr}$ is sensitive to $D_0$, when $D_0$ exceeds 1.5 mm. In addition, we identified considerable biases in the self-consistency W-band reflectivity evaluation (Myagkov et al., 2020). These biases likely result from overestimation of sensitivity of $\delta$ to $D_0$ and may reach 5 dB at $\delta$ smaller than $2°$. The empirical scattering model allows us to avoid these biases and to get the results consistent with the disdrometer-based approach.



The results of the current study provide a solid base for future work. First, the differences in $z_{dr}(D)$ and $\delta(D)$ between the SSA and ESM should be further explored. This may not only lead to an improved understanding of microphysical and scattering properties of raindrops but would also be helpful to simulate scattering properties in rain at arbitrary elevation angles and wavelengths of cloud radars. Second, as an intermediate solution, an empirical extension of the ESM to other elevation angles can be developed. This requires polarimetric observations at different elevation angles, which in general can

be done relatively soon taking into account the growing number of sites with polarimetric cloud radars. Third, in combination with the measurement error model from Myagkov and Ori (2022), the results of the current work can be used to develop an accurate DSD-profiling retrieval. An achievable accuracy is expected to be comparable to the existing dual-frequency approach (Tridon and Battaglia, 2015). The polarimetry-based method, however, potentially has advantages because only one radar unit is required and there are no strict requirements for pointing and synchronization in range and time as in the case of dual-

frequency approach. Fourth, the results of this work can be used for more accurate simulations of observables by a potential space-based polarimetric cloud radar (Battaglia et al., 2022). This can help to provide a more precise estimation of the added value of such a project.

*Code and data availability.*  The trained ANN for the ESM and low-turbulence spectra collected at locations 1–3 will be published in the supplementary materials upon acceptance of the manuscript.

**Appendix A: Spectral polarimetric products from complex spectra**

Elements of the spectral coherency matrix $\mathbf{B}(v_k)$ are calculated as follows:

$$B_{hh}(v_k) = \left\langle \dot{S}_h(v_k)\dot{S}_h^*(v_k) \right\rangle - N_h, \tag{A1}$$

$$B_{vv}(v_k) = \left\langle \dot{S}_v(v_k)\dot{S}_v^*(v_k) \right\rangle - N_v, \tag{A2}$$

$$\dot{B}_{hv}(v_k) = \dot{B}_{vh}(v_k)^* = \left\langle \dot{S}_h(v_k)\dot{S}_v^*(v_k) \right\rangle, \tag{A3}$$

where $\dot{S}_h$ and $\dot{S}_v$ are complex amplitudes measured in the horizontal and vertical channel, the dot indicates a complex value, $v_k = V_k/\sin\phi$ with $V_k$, $\phi$, and $k$ being the radial velocity, the elevation angle, and the index of a spectral line, respectively; the asterisk symbol denotes complex conjugation, $N_h$ and $N_v$ are noise levels estimated using the method from Hildebrand and Sekhon (1974) in the horizontal and vertical channel, respectively. The elements of $\mathbf{B}(v_k)$ are calibrated in linear radar-reflectivity units, i.e. mm$^6$ m$^{-3}$. The calculations are performed for each time sample and range bin. The radar reflectivity

spectra $Z_h(v_k)$ are equivalent to $B_{hh}(v_k)$.





Using the elements of $\mathbf{B}(v_k)$ we calculate spectral differential reflectivity $Z_{dr}(v_k)$, differential phase $\Phi_{dp}(v_k)$, and correlation coefficient $\rho_{hv}(v_k)$:

$$Z_{dr}(v_k) = \frac{B_{hh}(v_k)}{B_{vv}(v_k)}, \tag{A4}$$

$$\Phi_{dp}(v_k) = \arctan\left(\frac{\text{Im}\left[\dot{B}_{vh}(v_k)\right]}{\text{Re}\left[\dot{B}_{vh}(v_k)\right]}\right), \tag{A5}$$

$$\rho_{hv}(v_k) = \frac{\left|\dot{B}_{vh}(v_k)\right|}{\sqrt{B_{hh}(v_k)B_{vv}(v_k)}}, \tag{A6}$$

where Im and Re are imaginary and real parts of a complex number, respectively.

## Appendix B: Integrated variables

For calculation of integrated variables we use the normalized-gamma DSD (Illingworth and Blackman, 2002, Eq. 13 therein):

$$N(D) = \frac{0.033 N_L D_0^4 \Lambda^{\mu+4}}{\Gamma(\mu+4)} D^\mu \exp(-\Lambda D), \tag{B1}$$

where $\Lambda = (3.67 + \mu)/D_0$. We create a discreet vector of diameters from 0.01 to 5 mm with the step $\Delta D = 0.01$ mm. For each diameter bin $D_i$ a number of drops $N_i$ is calculated:

$$N_i = N(D_i)\Delta D, \tag{B2}$$

for given $N_L$, $\mu$, and $D_0$.

The integrated radar reflectivity and attenuation are calculated as follows:

$$z_h = \frac{10^{18}\lambda^4}{\pi^5|K|^2} \sum_{i=1}^{n} |\dot{s}_{hh}(D_i)|^2 N_i, \tag{B3}$$

$$A_h = 8.686 \times 10^3 \frac{2\pi}{k} \sum_{i=1}^{n} \text{Im}[\dot{f}_{hh}(D_i)] N_i, \tag{B4}$$

where $\lambda$ is the radar wavelength, $|K|^2$ is the dielectric factor of water, $k$ is the wave number, $\dot{s}$ and $\dot{f}$ are elements of the back- and forward-scattering matrix with the first and second subscripts indicating the polarization of the incident and scattered


waves, respectively. Polarimetric variables are obtained using the following equations:

$$z_{dr} = \frac{\sum_{i=1}^{n} |\dot{s}_{hh}(D_i)|^2 N_i}{\sum_{i=1}^{n} |\dot{s}_{vv}(D_i)|^2 N_i}, \tag{B5}$$

$$\delta = \frac{180}{\pi} \arg\left[\sum_{i=1}^{n} N_i \dot{s}_{hh}^*(D_i) \dot{s}_{vv}(D_i).\right], \tag{B6}$$

$$A_{dp} = 8.686 \times 10^3 \frac{2\pi}{k} \sum_{i=1}^{n} \mathrm{Im}[\dot{f}_{hh}(D_i) - \dot{f}_{vv}(D_i)] N_i, \tag{B7}$$

$$K_{dp} = 10^3 \frac{180}{\pi} \frac{2\pi}{k} \sum_{i=1}^{n} \mathrm{Re}[\dot{f}_{hh}(D_i) - \dot{f}_{vv}(D_i)] N_i. \tag{B8}$$

$\dot{s}_{hh}$, $\dot{s}_{vv}$, $\dot{f}_{hh}$, and $\dot{f}_{vv}$ are simulated using the SSA. The size-shape relation from Pruppacher and Pitter (1971) is used, if not stated otherwise.

In the case of the ESM:

$$z_{dr} = \frac{\sum_{i=1}^{n} |\dot{s}_{hh}(D_i)|^2 N_i}{\sum_{i=1}^{n} |\dot{s}_{hh}(D_i)|^2 N_i / \overline{z}_{dr}(D_i)}, \tag{B9}$$

$$\delta = \frac{180}{\pi} \arg\left[\sum_{i=1}^{n} N_i \sqrt{|\dot{s}_{hh}(D_i)|^4 / \overline{z}_{dr}(D_i)} e^{j\overline{\delta}(D_i)\pi/180}\right], \tag{B10}$$

where $j$ is the imaginary unit. $\overline{z}_{dr}(D_i)$ and $\overline{\delta}(D_i)$ are taken from the output of the approximation neural network discussed in Sec. 5.3.

*Author contributions.* AM and MF designed the experiment setup, AM and TN processed the data and prepared the paper with contribution from MF.

*Competing interests.* AM and TN are employees of Radiometer Physics GmbH. All authors have no other competing interests.

*Acknowledgements.* The authors are indebted to the staff of Hohenpeissenberg Observatory, the Agrion Foundation, the Institute of Sciences and Technologies for Sustainable Energy and Mobility (STEMS) of the Italian National Research Council, and the Polytechnic University of Turin, especially Dr. Mathias Gergely, Prof. Dr. Stefano Ferraris, Giorgio Capello, Dr. Marcella Biddoccu, Dr. Davide Canone, Alessio Gentile, and Simone Bussotti for the campaign planning, installation, deinstallation, and maintenance of the W-band radar. We also acknowledge the efforts of Chinese colleagues to install and maintain the radar in Shanghai. The work of AM and TN has been supported by RPG.





The W-band radar was provided by RPG. The authors thank RPG employees Jan Waßmuth and Philipp Schaffranek for the preparation

of the W-band radar for campaigns. The authors thank Prof. Dr. Alessandro Battaglia from the Polytechnic University of Turin for fruitful discussions on spectral polarimetry.





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
