# Peer review of "Empirical model for backscattering polarimetric variables in rain at W-band: motivation and implications"

_Atmospheric Measurement Techniques, 2024_

## Author Comment (AC1)

On behalf of all coauthors, we would like to thank the reviewer for the valuable comments on the manuscript. Below we address all the comments. Reviewer's comments are shown in blue color, while our responses are in black.

**Reviewer 1:**

**Comment:** Unclear what is the take home message in the remark at the end of Sect 4.2
**Response:** In the end of Sec 4.1 we mention that two additional sources of information are required to constrain the size-velocity relation. The finding from the disdrometer observations is one of these sources. We added the following sentence to the end of the section 4.2 "This finding is used in Sec.~\ref{sec:rad_dsd} to constrain the size-velocity relation, i.e. fitting parameters $a_{1,2,3}$."

**Comment:** Line 304: "because of smaller concentration of these drops and attenuation by liquid and gas" I disagree on the second reason, attenuation acts uniformly across the spectrum.
**Response:** We agree, our formulation is not precise enough. No doubt, the attenuation affects the entire spectrum. This, however, does not contradict the fact that attenuation is a factor pushing the observed spectral reflectivity for large drops below the scattering of a single drop. We changed the sentence "For drops larger than about 2~mm in diameter, the single drop reflectivity exceeds the median measured reflectivity because of small concentration of these drops."

**Comment:** Line 310-312: I have never noticed this secondary minimum. I would doubt these is due to non-spherical effects, I would be more in favor of considering DSD effects only (unless we really disproof Mie)
**Response:** From our measurements alone, we cannot judge which effect is responsible. At ERAD2024 we discussed this topic with several people. Some of them think that this is a resonance effect, others that this is a DSD effect. We wrote these two possible explanations, however, as is written in the end of the section 5.1, this topic is out of the scope of the manuscript. We extended the discussion of this topic in the section 5.1 as follows:
"One hypothesis is that this is due to scattering resonance caused by drop shapes diverging from the ideal spheroid. Another possible explanation is an increased concentration of drops with 0.7~mm due to a specific formation process. For example, \citet[][Sec.~15.5 therein]{Pruppacher1997} shows the formation of distinct peaks in equilibrium DSDs due to the processes of coalescence and breakup of droplets. Notably, the most prominent peak occurs within the sub-millimeter drop-size range. From our observations alone, however, we cannot conclude the exact reason of the maximum in observed spectra. Further laboratory and in situ-based investigations are required to answer this question. This is therefore out of the scope of the current study."

**Comment:** Not sure why Figure 9 comes before figure 8. Anyhow to me Fig9 is repetitive (you could include the red model lines in the left panels of fig.8
**Response:** The figure 8 is first discussed in the first sentence of the section 5.2, while the figure 9 is discussed in the last paragraph of the section 5.2. The sequence of figures is as expected, so we do not understand what the reviewer means in the first sentence of the comment. Addressing the second part of the comment, it is crucial to note that the two figures present different variables on the x-axis, namely diameter and terminal velocity. From our perspective, it is essential to include both figures. Figure 8 highlights the discrepancies between observed and simulated spectral polarimetric variables due to assumed size-shape relationships. In contrast, Figure 9 illustrates the differences that arise not only from assumed size-shape relationships but also from assumed size-velocity relationships. This comprehensive comparison underscores the multifaceted impact of these assumptions on our results.

**Comment:** 3: do we really need an artificial neural network? To me it just adds confusion. I would stick with a LUT based on Fig.8. Not sure what you add more than that.
**Response:** For various applications, such as variational retrievals, it is essential to have continuous and differentiable functions that approximate the derived dependencies of spectral polarimetric variables on drop diameter. As discussed at the beginning of Section 5.3, polynomial approximations may not be ideal due to the complex behavior of the functions being approximated. To address this, we utilize a tool commonly employed at our company for function approximation. We intend to share these results with the community along with

this manuscript upon acceptance. We will provide ready-to-use MATLAB functions, ensuring that readers do not need prior knowledge of neural network training to utilize these resources.

**Comment:** Sect 6.1: I see the differences between integrated ZDR and delta when using your ESM. Maybe it is worth comparing these differences with typical errors of such variable (you mention errors in delta, maybe it is worth also mentioning errors in zdr).

**Response:** Thank you for your suggestion. The uncertainties in ZDR and delta are determined by the quality of separation between backscattering and propagation polarimetric variables. For cloud radars, this separation is achieved using polarimetric spectra, as demonstrated by Myagkov et al. (2020) and Unal and van den Brule (2024). A significant advantage of this method is that the estimation of backscattering variables is immune to issues with polarimetric calibration and does not require prior knowledge of propagation effects. However, the quality of separation can vary from spectrum to spectrum due to factors such as air motions, integration time, range and Doppler resolution, and signal-to-noise ratio (SNR).

We can use the standard deviation of spectral delta and spectral ZDR in the Rayleigh part of the spectra as an initial estimate of the separation uncertainty. According to Figure 12 in Myagkov et al. (2020) and Table 2 in Unal and van den Brule (2024), the standard deviations for delta and ZDR in the Rayleigh part of the spectra within the first kilometer of observations do not exceed 0.3 degrees and 0.01 (in linear units), respectively. The differences between SSA and ESM reach 0.7 degrees and 0.3 dB (1.07 in linear units) in ZDR and delta, respectively. These differences are significant when compared to the uncertainties in the backscattering variables. This discussion has been added to the end of Section 6.1.

**Comment:** 11: what are the blue dots all exactly at 6 dB Z offset?

**Response:** The self-consistency method has several empirically chosen requirements for selecting profiles for calibration. When applied to a large dataset, there are instances where the method fails to converge to a correct result, resulting in outliers on the scatter plot. However, the percentage of these outliers is minimal. To enhance the visibility of the scatter plot, any offset produced by the method that exceeds 6 dB is capped at 6 dB. This information is now added to the caption of the Fig. 11.

Minor corrections:

**Comment:** Line 47 A compactness è The compactness
**Response:** corrected

**Comment:** Line 48: "A large number of cloud radars are capable of polarimetric measurements" (well there are few in the world, I would attenuate the statement.
**Response:** changed to "Cloud radars are often capable of polarimetric measurements."

**Comment:** Also statements at line 54-56 ( a bit vague, e.g. what do you mean with strong rain, I would rephrase them)
**Response:** Changed to "Due to attenuation by atmospheric gases and liquid water, cloud radars have spatial coverage orders of magnitude smaller than the coverage by operational centimeter-wavelength radars."

**Comment:** "an oscillatory behavior at drop sizes roughly proportional to half of the radar wavelength", there are multiple oscillations occurring at multiple size, rephrase
**Response:** Changed to "The simulated spectra of differential reflectivity $Z_{dr}$ show oscillations at drop sizes roughly proportional to half of the radar wavelength."

**Comment:** "In real rain measurements, however, we do see Zdr considerably exceeding 0.12 dB". It looks like a sentence out of the blue, not corroborated by any data or a reference. You need to explain more here or skip it. Also could that signal be caused by differential attenuation?
**Response:** This statement is based on our experience. In order to make this statement more solid, we introduced a subsection (6.2 in the updated version of the manuscript) demonstrating Zdr exceeding 0.12 deg. The sentence in the introduction we modified accordingly "However, as we demonstrate in this study, in real rain measurements, we do see $Z_{dr}$ considerably exceeding 0.12~dB."

**Comment:** Line 118-120. The radar actually provides spectra as a function of the radial velocity (V_k) not of v_k. A different thing is how you reprocess the data.

**Response:** We agree, the formulation was not clear enough. We modified the text as follows: "During operation the radar provides spectra of radar reflectivity $Z_h(V_k)$, differential reflectivity $Z_{dr}(V_k)$, differential phase $\Phi_{dp}(V_k)$, and correlation coefficient $\rho_{hv}(V_k)$, where $V_k$ is the radial velocity corresponding to the spectral component with index $k$. Within this study, we replace $V_k$ with $v_k = V_k / \sin \phi$ with $\phi$ being the elevation angle. In the absence of air movement, $v_k$ represents the terminal velocity of the droplets. The spectra are calculated as explained in Appendix~\ref{ap:pol_prod}."

**Comment:** Line 124: high è higher and lowè lower

**Response:** done

**Comment:** "still include observations in rain affected by strong attenuation", not sure how you can do that if Z<5 dBZ are excluded, or you need to specify what you mean with strong attenuation

**Response:** Please note that our focus is on a very short distance range, specifically within the first kilometer from the radar. As demonstrated by \citet{Hogan2003} and \citet{Matrosov2007}, the non-attenuated reflectivity at W-band exceeds 20~dBZ for rain rates exceeding 10~mm~h$^{-1}$. \cite{Aydin1991} estimates that at 100~mm~h$^{-1}$, the non-attenuated reflectivity reaches 33~dBZ. The one-way attenuation at 10 and 100~mm~h$^{-1}$ are 7 and 40~dB~km$^{-1}$, respectively \citep{Aydin1991,Matrosov2007}. Therefore, at a distance of 1 km, the attenuated reflectivity in 10~mm~h$^{-1}$ rain exceeds the used 5-dBZ threshold. At the minimum analyzed distance of 290~m, even observations in 100~mm~h$^{-1}$ rain fulfill the requirement. Since our processing is conducted on a single spectrum basis, it is not necessary to have complete profiles up to 1 km. This text is added to Section 3.1.

**Comment:** "all lines with SNR below 30 dB" è all spectral signal with ..... The term "lines" sounds a little bit ambiguous to me. Check its use.

**Response:** we changed the term "spectral line" to "spectral component" throughout the manuscript

**Comment:** Line 339: delete "a"

**Response:** done

**Comment:** Line 413: add spheroidal (before approximations)

**Response:** done

---

## Author Comment (AC2)

On behalf of all coauthors, we would like to thank the reviewer for the valuable comments on the manuscript. Below we address all the comments. Reviewer's comments are shown in blue color, while our responses are in black.

**Comment:** The authors select radar returns with strong SNRs (section 3.1) produced by significant rain. The range of radar observations can be longer than 1 km. An estimation of differential attenuation in rain would be desirable in analyzing the dual-pol variables. Differential attenuation decreases measured ZDR and δ values that could cause the decrease in these variables in Fig. 9.

**Response:** Thank you for your detailed comment. We have addressed each point separately below:

1. **Study Objective:** We would like to clarify that the primary aim of our study is not to build a retrieval or derive properties of rain. Instead, our goal is to select "ideal" spectra for analysis. We have collected several datasets and established a set of rules for selecting these "ideal" spectra, as described in Section 3. Therefore, there is no need to use data from distances greater than 1 km in this study.

2. **Differential Attenuation Estimation:** Estimating differential attenuation in rain is more straightforward for cloud radars compared to centimeter-wavelength radars. For cloud radars, we can use polarimetric spectra to separate backscattering and propagation effects. This separation approach is detailed in Section 3.3 of our paper, "AMT - Evaluation of the Reflectivity Calibration of W-band Radars Based on Observations in Rain." Additionally, the study "Exploring Millimeter-Wavelength Radar Capabilities for Raindrop Size Distribution Retrieval: Estimating Mass-Weighted Mean Diameter from the Differential Backscatter Phase" (Journal of Atmospheric and Oceanic Technology, Volume 41, Issue 6, 2024) covers spectra-based separation. Since these topics are already addressed in these publications, we have briefly discussed the separation in Section 3.2 of the current manuscript.

3. **Propagation Effects:** We agree that propagation effects can alter observed variables, such as ZDR and delta. However, we would like to emphasize that for cloud radars, we can separate propagation and backscattering effects using polarimetric spectra, as discussed in the aforementioned studies. We have applied these separation techniques to derive ZDR and delta values that are unaffected by propagation effects, as explained in Section 3.2 of the current manuscript. Consequently, Figure 9 presents results that have been corrected for propagation effects. We added the following text to the end of the Section 3.2: "We emphasize that propagation and hardware effects influence all spectral components uniformly within a given set of polarimetric spectra. These effects are accounted for in the biases $b_{dr}$ and $b_{\phi}$. Consequently, the estimates $z_{dr}(v_k)$ and $\delta(v_k)$ remain unaffected by these effects."

**Comment:** To my knowledge, the radar has an antenna radome. The radar measurements were taken in rain when the radome was wet. Water on radome affects dual-polarization measurements. If the authors have data on differential attenuation caused by a wet radome, they should be included in the manuscript.

**Response:** We agree, the problem of wet radomes is a crucial topic for precise polarimetric measurements. The used radar has an active rain mitigation system consisting of a hydrophobic material of radomes and a strong blower creating a narrow flow of air over the radome surface to blow water away. In addition, any issues with polarimetric calibration which may occur e.g. due to receiver drifts, affect all spectral components (small spherical drops and large drops responsible for ZDR and delta) in the same way. Therefore, the separation technique discussed in the previous response (and also in the Section 3.2 of the manuscript) allows us to estimate ZDR and delta unaffected by calibration issues. This is explained in the Section 3.2. We added the following text to the end of the Section 3.2: "We emphasize that propagation and hardware effects influence all spectral components uniformly within a given set of polarimetric spectra. These effects are accounted for in the biases $b_{dr}$ and $b_{\phi}$. Consequently, the estimates $z_{dr}(v_k)$ and $\delta(v_k)$ remain unaffected by these effects."

**Comment:** The authors compare radar data with data obtained from the Thies disdrometer located within 10 m of the radar (p. 5, Ln. 144). At a slant distance of radar observation of 1000 m (Table 2) and 30 deg of antenna elevation, the height of radar resolution volume is 500 m above the radar position and about 860 m in the horizontal direction. That is, an assumption of the uniformity of rain at those distances is made. Justification is needed.

**Response:** Please note that disdrometer data serve two purposes in this study. First, they are used to constrain the size-velocity relations of raindrops in Section 4. For this purpose, we use disdrometer data that are neither collocated with the radar nor cover the same time periods (see Section 2.2). The rationale is that size-velocity relations are expected to be consistent regardless of the location and time of measurement. We focus on characterizing drops on average rather than analyzing them at specific moments. The effect of air density, which depends on atmospheric pressure, is taken into account.

Second, collocated disdrometer data (see Section 2.3) are used to evaluate radar reflectivity in Section 6.3 of the updated manuscript. In this case, the distance between the radar range bin and the disdrometer is 300 m. We follow the calibration approach detailed in Section 4 of AMT - Evaluation of the reflectivity calibration of W-band radars based on observations in rain. The effects of evaporation and time lags are considered. The narrow distribution of results (within +/-1 dB) indicates that horizontal and vertical displacements have no significant impact. We added the following text to the section 6.3. of the updated manuscript "The narrow distribution of the results suggests that any potential effects from the horizontal displacement of the radar's range bin and the disdrometer fall within the method's uncertainty."

**Comment:** The authors indicate that their constraints are meant to avoid turbulent areas (section 3). Rain frequently suppresses turbulence, but the horizontal wind shear can be significant. The effectiveness of raindrops as wind tracers depends on their sizes. The horizontal wind shear reshuffles raindrops and its impact on radar spectra can be significant. Therefore, information on possible wind shears would be informative. Such information can be obtained from the collected radar data by analyzing the Doppler velocity along the radials.

**Response:** We would like to clarify that the set of rules introduced in Section 3 for selecting spectra were specifically designed to exclude those affected by turbulence, regardless of its source. The reviewer's comment is not entirely clear to us. Our study does not aim to characterize all spectra; instead, we focus on selecting and processing only "ideal" spectra that have minimal or no turbulence effects.

**Comment:** I am curious why eq. (3) has the tanh(.) functions? Is there any reason for that? The curve in Fig 6a is quite smooth to be approximated with a simple polynomial function as it is typically done.

**Response:** The choice of approximation approach is always a matter of preference. As mentioned in Section 4.1, we use the hyperbolic tangent function primarily because it offers sufficient flexibility to fit the size-velocity relations with just three parameters, and it ensures that the approximation is monotonic, which is not always the case with polynomials. While other approximation methods could also fit the data, they would not change the results. We have provided the coefficients and the parameterization in Equation 3, allowing readers to easily use the results.